# Expression of Acetabular Labral Vascular Endothelial Growth Factor and Nerve Growth Factor Is Directly Associated with Hip Osteoarthritis Pain: Investigation by Immunohistochemical Staining

**DOI:** 10.3390/ijms24032926

**Published:** 2023-02-02

**Authors:** Yoshihiro Sato, Tomonori Tetsunaga, Kazuki Yamada, Yoshi Kawamura, Aki Yoshida, Toshifumi Ozaki

**Affiliations:** 1Department of Orthopaedic Surgery, Graduate School of Medicine, Dentistry and Pharmaceutical Sciences, Okayama University, Okayama 700-8558, Japan; 2Department of Intelligent Orthopaedic System, Faculty of Medicine, Dentistry and Pharmaceutical Sciences, Okayama University, Okayama 700-8558, Japan; 3Department of Medical Materials for Musculoskeletal Reconstruction, Graduate School of Medicine, Dentistry and Pharmaceutical Sciences, Okayama University, Okayama 700-8558, Japan

**Keywords:** hip osteoarthritis (OA), acetabular labrum, vascular endothelial growth factor (VEGF), nerve growth factor (NGF), immunochemical staining

## Abstract

The acetabular labrum enhances hip joint stability and plays a key role in osteoarthritis (OA) progression. Labral nerve endings contribute to hip OA pain. Moreover, vascular endothelial growth factor (VEGF) and nerve growth factor (NGF) are associated with pain. Consequently, we analysed VEGF and NGF expression levels in the labrum and their roles in OA. Labra obtained from OA patients were stained immunohistochemically, and labral cells were cultured and subjected to a reverse transcription (RT)–polymerase chain reaction (PCR) to analyse *VEGF* and *NGF* mRNA expression. VEGF and NGF expression were compared in each region of the labrum. Correlations between VEGF and NGF expression and age, body mass index, Kellgren–Lawrence grade, Harris Hip Score, the visual analogue scale (VAS), and Krenn score were analysed, and the RT-PCR confirmed the findings. VEGF and NGF expression were high on the labral articular side, negatively correlated with the Krenn score, and positively correlated with the VAS in early OA. *VEGF* and *NGF* mRNA expression increased significantly in patients with severe pain and decreased significantly in severely degenerated labra. In early OA, VEGF and NGF expression in the acetabular labrum was associated with the occurrence of hip pain; therefore, these factors could be effective targets for pain management.

## 1. Introduction

The acetabular labrum, a fibrocartilaginous ring that lines the acetabular margin, increases acetabular depth and enhances joint stability. The structure contributes to joint fluid retention at the articular surface, increasing joint stability [1]. Acetabular labrum damage is positively correlated with osteoarthritis (OA) progression [2].

The development of new analgesics to treat OA-associated pain is a challenge. In a review of the literature on the relationship between the gut microbiome and OA pain, Sánchez-Romero et al. [3] reported that studies have begun focusing on the mechanisms underlying OA pain. Therapies that target the acetabular labrum, a tissue of the hip joint, are important for alleviating OA-associated pain. A better understanding of the pathogenesis of labral degeneration and the mechanisms associated with OA would facilitate the development of advanced treatment strategies for OA.

Although the association between vascular endothelial growth factor (VEGF) expression and OA has been reported [4,5], the relationship between VEGF expression and acetabular labrum degeneration remains unclear. Given that labral vascular flow is related to VEGF expression, investigating VEGF expression in the labrum is crucial for elucidating its role in the degenerative pathogenesis of OA. The acetabular labrum is abundant in free nerve endings and sensory nerves that are implicated in hip pain [6,7,8]. Nerve growth factor (NGF) is involved in nerve development [9]. Studies have shown that angiogenesis contributes to pain in OA patients [10,11]. Some studies have suggested an association between VEGF and NGF expression and pain using animal models [12,13,14,15]. Furthermore, VEGF expression in the synovial membranes of patients with painful knee OA has been confirmed [16]. However, previous studies have not investigated VEGF and NGF expression in the acetabular labra of patients with painful hip OA.

The purpose of this study was to investigate VEGF and NGF expression levels in the labrum and their role in OA. Based on our experience with patients suffering from severe pain despite relatively mild hip deformities, we hypothesised that a putative association occurred between VEGF and NGF expression in the acetabular labrum and the occurrence of hip pain. Thus, VEGF and NGF expression and localisation and factors that influence their expression were analysed to elucidate their association with hip pain. The findings of this study may offer insights into the mechanisms underlying hip pain in patients with hip OA.

## 2. Results

Forty-four acetabular labral samples were obtained from patients who underwent total hip arthroplasty (THA) for hip OA at our hospital. The samples were investigated histologically and immunohistochemically. Further details are provided in the Materials and Methods section.

### 2.1. Histological Analysis of the Acetabular Labrum 

A histological examination of the 44 specimens revealed no vascular structures on the labral surface facing the femoral head, whereas 35 of the 44 specimens showed vascular structures on the surface facing the articular capsule. Labral degeneration was evaluated using the Krenn score: 10 specimens were scored 0, 13 were scored 1, 14 were scored 2, and 7 were scored 3.

### 2.2. Immunohistochemistry of VEGF and NGF

The outer region was more vascularised than the inner region. In the inner region, the number of cells was low, but the proportion of antibodies was high. In the outer region, the number of cells was high, but the proportion of antibodies was low. Similar results were obtained for VEGF and NGF (Figure 1).

A comparison of VEGF and NGF expression revealed a significantly lower number of VEGF- and NGF-positive cells in the labral inner region than in the labral outer region (*p* < 0.05; Figure 2).

### 2.3. Correlations of Inner VEGF and NGF Levels with Age, Gender, BMI, KL Grade, HHS, the VAS, and Krenn Score

Spearman’s correlation analysis demonstrated that the positive staining rate of the inner VEGF was not significantly correlated with age (*p* = 0.10), body mass index (BMI; *p* = 0.097), Kellgren–Lawrence (KL) grade (*p* = 0.99), Harris Hip Score (HHS; *p* = 0.73), or the visual analogue scale (VAS; *p* = 0.71; Figure 3). Furthermore, the positive staining rate of the inner VEGF was negatively correlated with the Krenn score (r = −0.83, *p* < 0.001). Similar correlations were observed in the positive staining rate of the outer VEGF, which was negatively correlated with the Krenn score (r = −0.77, *p* < 0.001). The *t*-test results showed no significant difference in inner VEGF expression between males and females (*p* = 0.25).

The positive staining rate of the inner NGF was not significantly correlated with age (*p* = 0.18), BMI (*p* = 0.45), KL grade (*p* = 0.21), HHS (*p* = 0.20), or the VAS (*p* = 0.71; Figure 4), and it was negatively correlated with the Krenn score (r = −0.83; *p* < 0.001). Similarly, the positive staining rate of the outer NGF was negatively correlated with the Krenn score (r = −0.72; *p* < 0.001). The *t*-test results showed no significant difference in inner NGF expression between males and females (*p* = 0.072).

As similar correlations were observed in the inner and outer regions, the results discussed in the succeeding sections are focused on the inner region.

### 2.4. VEGF and NGF Expression Is Correlated with Pain in Early OA

Because bone pain strongly affects patients with severe joint deformity, the patients were divided into two groups: early and late OA groups. Correlations between the inner VEGF and NGF levels and the VAS were examined. Correlations between the inner VEGF level and VAS (r = 0.64; *p* = 0.0088; Figure 5a) and between the inner NGF level and VAS (r = 0.61; *p* = 0.014; Figure 5c) were observed in the 15 specimens with KL grades 0–2. However, correlations between either the inner VEGF and VAS (r = 0.062; *p* = 0.74; Figure 5b) or the inner NGF and VAS (r = −0.21; *p* = 0.28; Figure 5d) were not observed for the 29 specimens with KL grades 3–4.

### 2.5. VEGF and NGF mRNA Expression Significantly Decreases with the Progression of Acetabular Labral Degeneration

A comparison of the Δ cycle threshold (Ct) results revealed significantly lower VEGF mRNA expression in patients with severe degeneration than in patients with mild degeneration (*p* = 0.042; Figure 6a). Similar results were obtained for NGF (*p* = 0.012; Figure 6b).

### 2.6. VEGF and NGF mRNA Expression Significantly Increases in Patients with Severe Pain

A comparison of the ΔCt results revealed significantly higher VEGF and NGF mRNA expression in patients with severe pain than in patients with mild pain (*p* = 0.0043; Figure 6c). Similar results were obtained for NGF (*p* = 0.0087; Figure 6d). 

## 3. Discussion

This study demonstrated high VEGF and NGF expression on the acetabular labral surface facing the femoral head, which was downregulated with the progression of labral degeneration. Patients with early OA who exhibited higher acetabular labral VEGF and NGF expression also complained of more severe pain. Some studies have explored the acetabular labrum in hip OA [17,18,19]. To the best of our knowledge, this is the first study on the relationship between VEGF and NGF expression in the acetabular labrum and pain.

The acetabular labrum presents a layered structure with different site-specific characteristics. Petersen et al. [1] reported the structure of the acetabular labrum based on electron microscopic observations of the labrum tissue microstructure. The acetabular labral structure differs between the femoral head and articular capsule, with evidence of vascular structures present in only one-third of the latter. Kelly et al. [20] examined the distribution of blood flow in cadaveric specimens and found that labral blood flow was abundant on the articular capsule side. Here, a histological analysis revealed cross-sections of vascular structures on the labral surface facing the articular capsule, and they were absent on the femoral head side. In contrast, VEGF expression, which is known to promote angiogenesis, was significantly higher on the femoral head side. Seldes et al. [21] reported the association of labral tears with increased microvascularity in the labrum located at the base of the tear and adjacent to the labral attachment to the bone. Hence, VEGF is considered to promote healing by inducing angiogenesis in the injured area of the acetabular labrum.

Nerve endings in the acetabular labrum are located on the femoral head side. Alzaharani et al. [22] reported changes in nerve distribution with OA progression, including a higher concentration of free nerve endings on the femoral head side than that on the articular capsular side. Kapetanakis et al. [23] reported a significant difference in the acetabular labral histological architecture between healthy individuals and patients with radiological grade III and IV hip OA, in which the conversion of mechanical receptors to free nerve endings resulted in disease progression and classical clinical manifestations. Here, we found higher NGF expression on the femoral head side in OA labra, which may lead to neuronal proliferation. Our results are consistent with those of previous studies on neural distribution [22,23]. Given that NGF induces the stretching of acetabular labral nerve endings [9], its expression may be strongly associated with hip pain that originates from the acetabular labrum. 

We observed a negative correlation between acetabular labral VEGF- and NGF-positive cell ratios and Krenn scores. This was supported by the RT-PCR results, which demonstrated significantly lower *VEGF* and *NGF* expression in acetabular labra with Krenn scores of 2 or 3 than in acetabular labra with Krenn scores of 0 or 1.

A histological and immunohistochemical study on acetabular labral degeneration with samples harvested during THA revealed a heterogenic matrix in all cases [24]. A portion of these cases exhibited calcification (17/77), angiogenesis (30/77), and macrophage infiltration into damaged areas during labral healing (23/77). A comparative analysis of the expression of the genes involved in labral angiogenesis between patients with femoroacetabular impingement and those with OA revealed significantly lower *VEGF* expression in the latter [25]. These findings suggest that a reduced acetabular labral angiogenic ability results in a concomitant reduction in healing, leading to the progression of OA. Here, we demonstrated a negative correlation between *VEGF* and *NGF* expression and the degree of labral degeneration and thus conclude that *VEGF* and *NGF* expression induces angiogenesis and neuronal proliferation in the acetabular labra of OA patients and that the progression of labral degeneration, accompanied by a decrease in *VEGF* and *NGF* expression, leads to OA worsening.

Sánchez-Romero et al. reported that the most visible changes after hip surgery in systemic biological markers, such as increased platelet and C-reactive protein concentrations, occur during days 1 to 3 postoperatively, and these may have an effect on some of the clinical outcomes, such as oedema, pyrexia, and postoperative pain that develop after surgery [26]. THA impacts patients in early postoperative stages in a similar way from a systemic point of view. This can help clinicians plan their postoperative management and better understand the postoperative course after THA. However, it is very difficult to objectively evaluate pain before surgery. Therefore, we investigated VEGF and NGF expression levels in the labrum and their role in OA. VEGF and NGF are reportedly associated with pain. VEGF-induced neovascularisation is known to promote the infiltration of inflammatory cells into periarticular tissues, whereas NGF promotes nerve outgrowth and increases pain sensitivity [10,11,12,13,14,15]. Takano et al. [16] reported a correlation between the VAS and VEGF expression in synovial tissue collected from patients with knee OA and significantly higher VEGF expression in patients with severe pain. Sánchez-Romero et al. [26] compared the postoperative blood test results of patients who underwent total hip and knee arthroplasty and reported that they did not significantly differ; thus, we decided that studies of the knee joint could be used as a reference for our study of the hip joint. Similarly, Ohashi et al. [27] reported the correlation between synovial NGF expression and the degree of pain and central sensitisation in patients with hip OA. Here, acetabular labral VEGF- and NGF-positive cell ratios were correlated with the VAS in patients with early OA. In patients with hip OA, the RT-PCR analysis revealed significantly higher labral *VEGF* and *NGF* mRNA expression in patients with VAS values ≥ 70 mm than in those with VAS values < 70 mm. These findings confirmed the correlation of acetabular labral VEGF and NGF expression with pain.

We considered the following relationship between VEGF and NGF expression in the acetabular labrum and pain. The initial processes in the mechanism underlying pain associated with OA involve VEGF and NGF expression in response to acetabular labral damage in early OA. This results in angiogenesis induction and nerve cell proliferation, leading to hip pain in patients with minimal deformities. With the progression of acetabular labral degeneration, joint deformities develop due to the worsening of OA. This stage is characterised by low VEGF and NGF expression in the degenerated acetabular labrum and decreased hip pain. 

This study has certain limitations. First, this study involved an analysis of specimens obtained only from patients with severe pain who were indicated for surgery, considering the difficulties associated with ethically harvesting healthy labral specimens, and this resulted in higher VAS scores in our patient cohort. Furthermore, hip pain may be due to damage to a multitude of structures, including the synovium or subchondral bone, not only the labrum.

Future directions: A comprehensive analysis would require an investigation of acetabular labral samples from patients without a history of hip pain, such as those collected from surgical cases of a femoral neck fracture. Behavioural analysis should be used to evaluate pain in acetabular labral VEGF- and NGF-overexpressing mice. Additionally, investigating OA development in VEGF/NGF-inhibited or -knockdown mice may provide deeper insights into the correlation between VEGF/NGF and pain associated with hip OA. Findings from further research may provide insights into the mechanisms underlying hip pain in patients with hip OA and lead to the development of more effective pain control treatments as compared to those currently available. The establishment of therapies targeting VEGF and NGF, which can alleviate hip pain and prevent labral degeneration, would be a breakthrough that could relieve patient suffering and halt the progression of OA.

## 4. Materials and Methods

VEGF and NGF expression and localisation in the labrum and factors that influence their expression were investigated by immunohistochemically staining the acetabular labrum, and the results were confirmed by RT-PCR.

### 4.1. Tissue Collection and Preparation

Forty-four acetabular labral samples were obtained from 12 male and 32 female patients who underwent THA for hip OA between April 2018 and December 2019 at our hospital (Okayama University Hospital, Okayama, Japan). All patients were diagnosed with primary osteoarthritis of the hip, and patients suffering from hip–spine syndrome were excluded from this study. The patients’ mean age at the time of surgery was 60.7 (range: 40–85) years and the BMI was 25.7 (range: 17.0–37.3) kg/m^2^ (Table 1). Cartilage degeneration was evaluated by three blinded assessors, who were expert hip surgeons, and was graded according to the KL classification as follows [28]: 1 hip was graded 0, 6 hips were graded 1, 8 hips were graded 2, 16 hips were graded 3, and 13 hips were graded 4. The mean HHS was 55.6 (range: 20–96) points, and the mean VAS value was 77.5 (range: 16–100) mm. They were measured from the patients’ subjective preoperative complaints.

### 4.2. Cells and Cell Culture

Each specimen was sectioned radially from the thickest part of the acetabular labrum perpendicular to the articular surface, and 3 mm thick slices were obtained. Each slice was sectioned into two; one slice was used for the histological examination and the other for the cell culture. The specimens for cell culture were finely engraved and cultured in low-glucose Dulbecco’s modified Eagle’s medium (Sigma-Aldrich, St. Louis, MO, USA) supplemented with 10% foetal bovine serum (HyClone, South Logan, UT, USA) and 1% penicillin/streptomycin (Sigma-Aldrich). Adherent cells were cultured at a density of 2500 cells/cm^2^ on type 1 collagen-coated polystyrene tissue culture dishes (Iwaki, Shizuoka, Japan) at 37 °C in a humidified atmosphere containing 5% CO_2_. The medium was changed every 3 d, and cells between passages 1 and 2 were used in further experiments.

### 4.3. Histological Examinations

Acetabular labral samples were sliced to 3 mm thick specimens and were preserved for 2 d in 10% buffered formalin. Further, the specimens were embedded in paraffin and cut into 4 µm thick sections.

Haematoxylin and eosin staining was performed for histological analysis, and acetabular labrum degeneration was evaluated using the Krenn score for fibrocartilage degeneration in the meniscus and labrum (grades 0–3; Table 2) [17,18,29] by three blinded assessors who were expert hip surgeons.

### 4.4. Immunohistochemical Staining

Specimens that were serial sections of those used for histological examinations were blocked with 3% skimmed milk in phosphate-buffered saline (PBS) with 0.3% Triton X-100 at room temperature (15–25 °C) for 90 min. Immunohistochemical staining was performed with the following antibodies: anti-VEGF (1:100; Santa Cruz Biotechnology Inc., Santa Cruz, CA, USA), a general marker of the vascular endothelium [4,5], and anti-NGF (1:50; Santa Cruz Biotechnology Inc.), which plays an important role in the development of sensory neurons responsible for nociception and temperature sensation [9]. The sections were incubated with primary antibodies at 4 °C in a moisture chamber overnight and then with mouse IgGk BP-HRP (1:25; Santa Cruz Biotechnology Inc.) at 25 °C for 60 min. After each step, the sections were washed twice with PBS. The sections were then washed and incubated with 3,3′-diaminobenzidine for 5–20 min, followed by extensive washing with distilled water. Finally, the nuclei were counterstained with haematoxylin.

The tissue sections were viewed under a microscope (BX53; Olympus, Tokyo, Japan), and images were captured using the image processing software cellSens standard version 1.18 (Olympus). Three images were captured at standardised locations on each section, namely, the inner region on the labral surface facing the femoral head and the outer region on the labral surface facing the joint capsule. VEGF- and NGF-positive cells, as well as all cells in the images in the inner region, were counted to calculate the percentages of positively stained cells [19], which were referred to as inner VEGF and inner NGF levels, respectively. Similarly, the percentages of positive cells in the outer region were referred to as outer VEGF and outer NGF levels.

### 4.5. Reverse Transcription (RT)–Polymerase Chain Reaction (PCR) and Quantitative Real-Time PCR Analyses

Total RNA was isolated from cultured labrum cells using ISOGEN reagent (Nippon Gene, Toyama, Japan). RNA samples (200 ng) were reverse transcribed to cDNA using a PrimeScript RT reagent kit (Takara Bio, Shiga, Japan). A real-time PCR was performed on the AriaMx real-time PCR system (Agilent Technologies, Santa Clara, CA, USA) using the TaqMan^®^ gene expression assay kit (Thermo Fisher Scientific, Waltham, MA, USA) and the primers listed in Table 3.

Each sample was subjected to PCR in triplicate under the following thermocycling conditions: denaturation at 95 °C for 10 min, followed by 40 cycles comprising denaturation at 95 °C for 15 s and annealing and elongation at 60 °C for 1 min. The cycle number at which the PCR product was first detected above a fixed threshold (cycle threshold) was determined for each sample. Changes in the target gene expression were calculated using ∆Ct as follows:$$\Delta \mathrm{Ct}={\mathrm{Ct}}^{\mathrm{target}}-{\mathrm{Ct}}^{\mathrm{GAPDH}}$$

### 4.6. Statistical Analysis

The results are presented as the mean ± standard error of the mean. Comparisons between groups were performed using a two-way paired Student’s *t*-test. Correlations between inner VEGF and NGF levels and age, BMI, KL grade, HHS, the VAS, and Krenn score in all 44 cases were analysed using Spearman’s correlation analysis. As we considered that bone pain strongly affects pain in patients with severe joint deformity, patients were divided into two groups; as the mean KL grade was 2.82, patients were divided into early (KL grade 0–2) and late (KL grades 3–4) OA groups. The correlation of inner VEGF and NGF levels with the VAS was examined using Spearman’s correlation coefficient in 15 specimens from patients with early OA and 29 specimens from patients with late OA. In RT-PCR analysis, a comparison of ΔCt was performed between patients with mild (Krenn scores of 0 or 1) and severe (Krenn scores of 2 or 3) degeneration. As the mean value of the VAS was 77.5 mm, patients were divided into two groups: patients with mild (VAS < 70 mm) and severe (VAS ≥ 70 mm) pain. ΔCt was compared between patients with mild pain and those with severe pain. All statistical analyses were performed using IBM SPSS Statistics 27 (IBM Corp., Armonk, NY, USA). The significance level was set at *p* < 0.05.

## 5. Conclusions

Our findings demonstrate high VEGF and NGF expression on the acetabular labral surface facing the femoral head. *VEGF* and *NGF* expression were downregulated with labral degeneration. Furthermore, in early OA, higher *VEGF* and *NGF* expressions on the acetabular labrum were associated with higher levels of pain. Studies on the correlations among the genes involved in cartilage metabolism, osteogenesis, and angiogenesis are essential for better comprehension of hip OA pathophysiology.

## Figures and Tables

**Figure 1 ijms-24-02926-f001:**
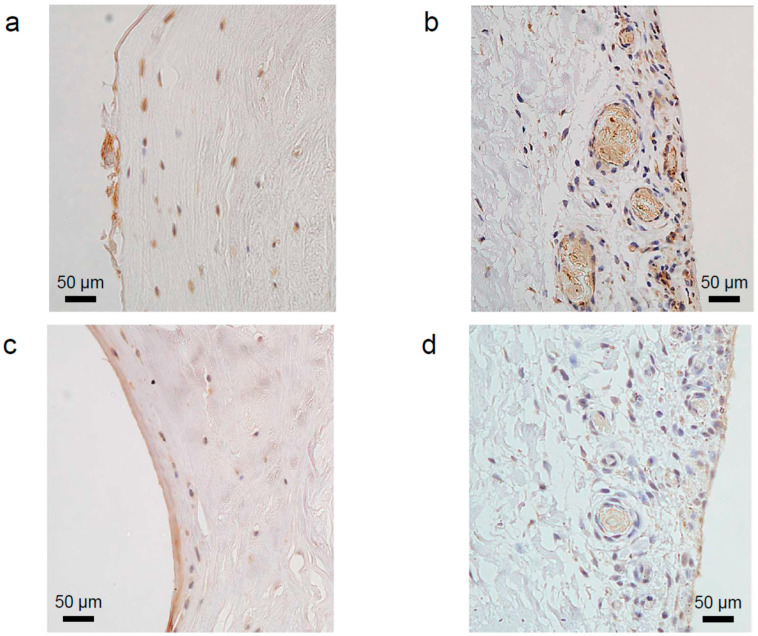
Immunohistochemical staining of vascular endothelial growth factor (VEGF) on the labral surface facing the femoral head (**a**) and the articular capsule (**b**). Immunohistochemical staining of nerve growth factor (NGF) on the labral surface facing the femoral head (**c**) and the articular capsule (**d**). The characteristics of staining in each region showed that the number of cells in the inner region was lower than that in the outer region. In the outer region, only some cells were stained positively. Vascular structures were observed in the labral outer region. Bars represent 50 µm (original magnification, 200×). The nuclei were counterstained with haematoxylin.

**Figure 2 ijms-24-02926-f002:**
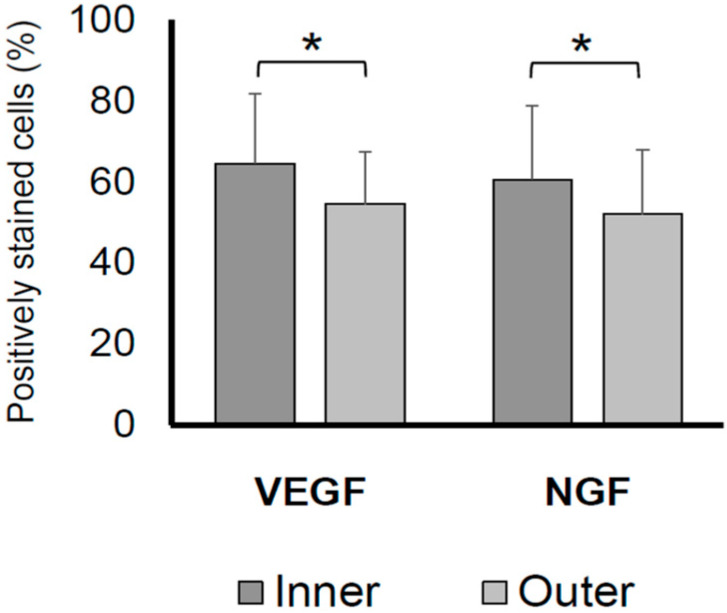
Percentages of VEGF- and NGF-positive cells on the inner and outer acetabular labral surfaces. The VEGF- and NGF-positive stained cell rate in the labral inner region was significantly higher than that in the labral outer region (*p* < 0.05). * *p* < 0.05. NGF, nerve growth factor; VEGF, vascular endothelial growth factor.

**Figure 3 ijms-24-02926-f003:**
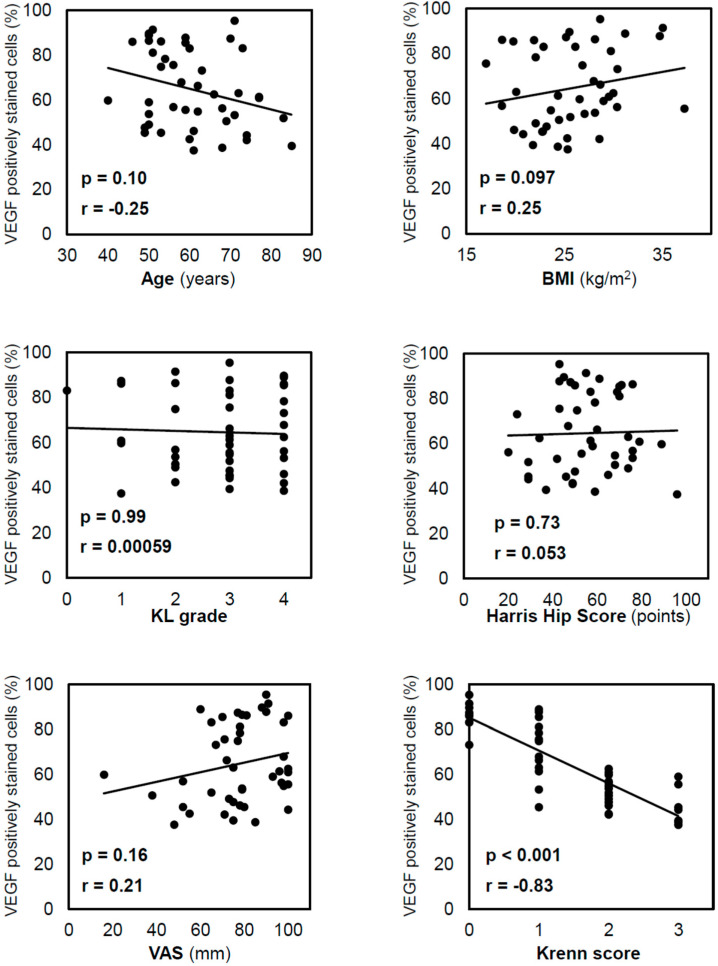
Correlation between VEGF-positive staining in cells in the inner region and other variables. Correlations were analysed using Spearman’s correlation coefficient. VEGF expression decreased with the progression of acetabular labral degeneration, as reflected by the Krenn score. VEGF, vascular endothelial growth factor; BMI, body mass index; KL, Kellgren–Lawrence; VAS, visual analogue scale.

**Figure 4 ijms-24-02926-f004:**
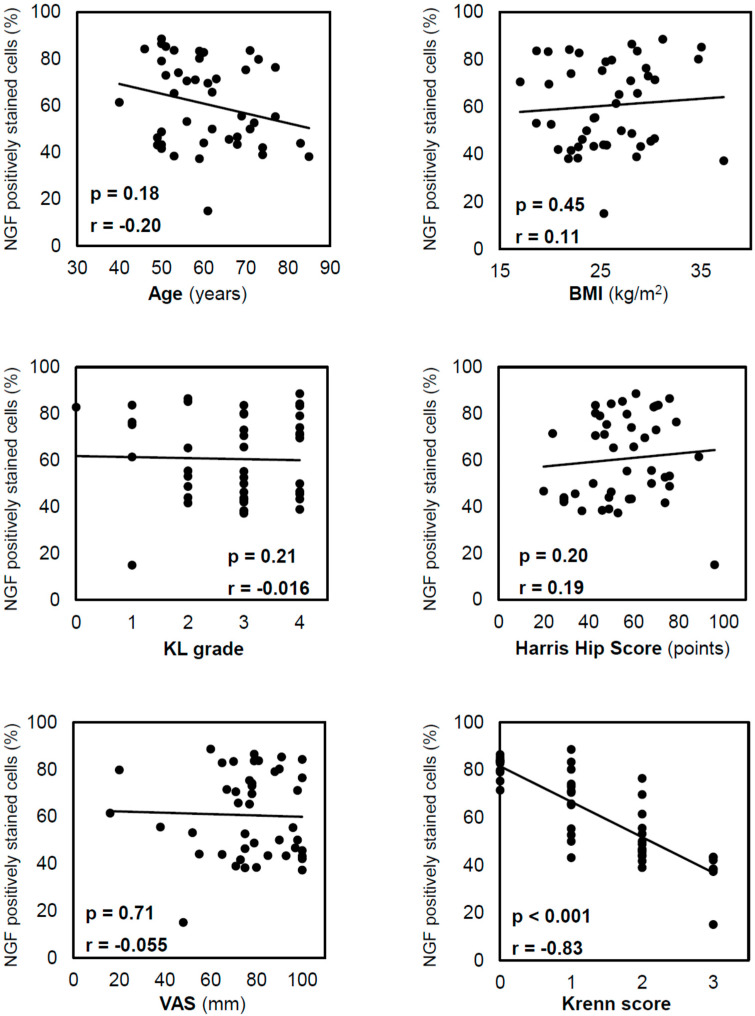
Correlation between NGF-positive staining in cells in the inner region and other variables. Correlations were analysed using Spearman’s correlation coefficient. NGF expression decreased with the progression of acetabular labral degeneration, as reflected by the Krenn score. NGF, nerve growth factor; BMI, body mass index; KL, Kellgren–Lawrence; VAS, visual analogue scale.

**Figure 5 ijms-24-02926-f005:**
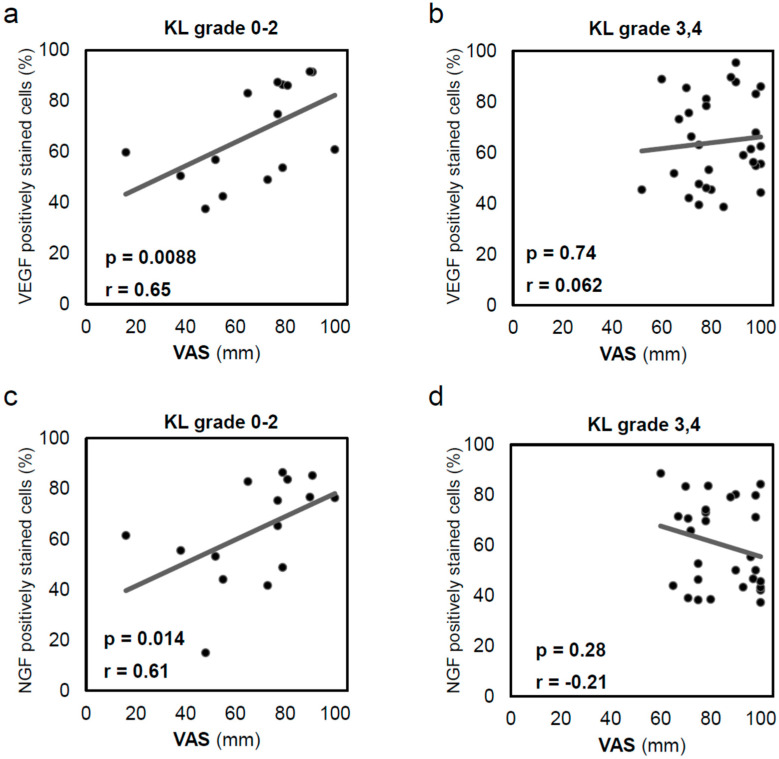
Correlation between inner acetabular labral VEGF expression and visual analogue scale (VAS) in patients with (**a**) Kellgren–Lawrence (KL) grade 0–2 OA (n = 15) and (**b**) KL grade 3–4 OA (n = 29). In early OA, patients reported more severe pain as VEGF expression increased. Correlation between inner acetabular labral NGF expression and VAS in patients with (**c**) KL grade 0–2 OA (n = 15) and (**d**) KL grade 3–4 OA (n = 29). In early OA, patients reported more severe pain as NGF expression increased. Correlations were analysed using Spearman’s correlation coefficient. VEGF, vascular endothelial growth factor; NGF, nerve growth factor; KL, Kellgren–Lawrence; VAS, visual analogue scale.

**Figure 6 ijms-24-02926-f006:**
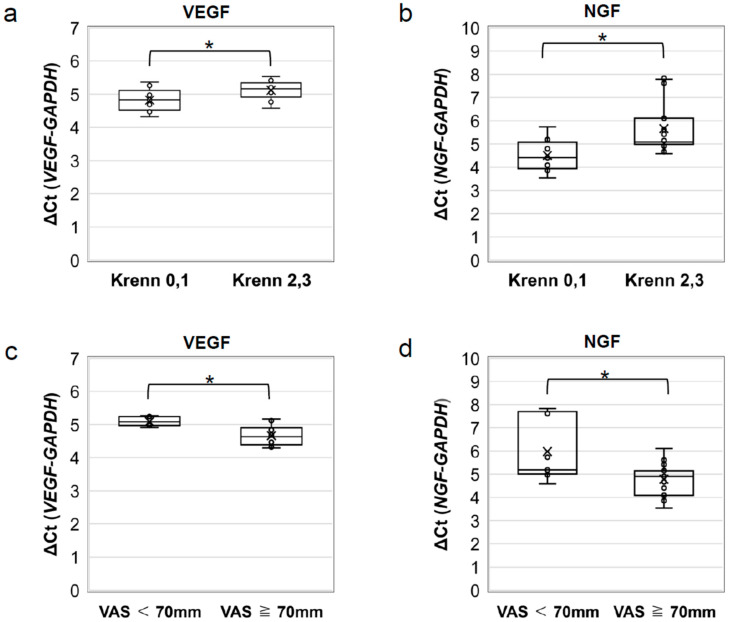
RT-PCR analysis of *VEGF* and *NGF* mRNA expression. Comparison of the ΔCt of (**a**) *VEGF* and (**b**) *NGF* mRNA expression between patients with mild (Krenn score 0 or 1) and severe (Krenn score 2 or 3) degeneration. *VEGF* and *NGF* mRNA expression was found to be lower in the group with severe acetabular labral degeneration. Comparison of ΔCt of (**c**) *VEGF* and (**d**) *NGF* mRNA expression between patients with mild or moderate pain (VAS < 70 mm) and severe pain (VAS ≥ 70 mm). *VEGF* and *NGF* mRNA expression was higher in the group in which patients reported severe pain. * *p* < 0.05. GAPDH, glyceraldehyde-3-phosphate dehydrogenase; NGF, nerve growth factor; VAS, visual analogue scale; VEGF, vascular endothelial growth factor.

**Table 1 ijms-24-02926-t001:** Patients’ clinical assessment data (n = 44).

Characteristic	Value
Sex, male/female	12/32
Age (years)	60.7 ± 10.6
BMI (kg/m^2^)	25.7 ± 4.5
KL classification	
Grade 0	1
Grade 1	6
Grade 2	8
Grade 3	16
Grade 4	13
HHS (points)	55.6 ± 17.3
VAS (mm)	77.5 ± 18.3

All data are reported as the mean ± standard deviation. BMI: body mass index; KL: Kellgren–Lawrence; HHS: Harris Hip Score; VAS: visual analogue scale.

**Table 2 ijms-24-02926-t002:** Krenn score: histopathological evaluation of the grade of fibrocartilage degeneration (0–3).

Grade	Features
0	Normal histological morphology
	Isomorphic chondrocytes
	Homogeneous eosinophil-stained matrix
	Regular cellularity
1	Low-grade degeneration
	Small reduction in cellularity (small areas)
	Nonhomogeneously stained matrix
	Small fissures in the matrix
2	Moderate degeneration
	Moderate reduction in cellularity (large areas)
	Variable size and shape of chondrocytes
	Moderate fissures in the matrix
3	High-grade degeneration
	Marked reduction in cellularity
	Large areas of complete chondrocytes loss
	Reticular/basophilic stained matrix (mucoid degeneration)
	Large fissures in the matrix (pseudocysts)

**Table 3 ijms-24-02926-t003:** Reverse transcription PCR primers.

	Gene	ID Numbers
*VEGFA*	Vascular endothelial growth factor A	Hs00911700_m1
*NGF*	Nerve growth factor	Hs00913377_m1
*GAPDH*	Glyceraldehyde-3-phosphate dehydrogenase	Hs03929097_g1

PCR, polymerase chain reaction.

## Data Availability

Not applicable.

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
