# Peer review of "Expression of Acetabular Labral Vascular Endothelial Growth Factor and Nerve Growth Factor Is Directly Associated with Hip Osteoarthritis Pain: Investigation by Immunohistochemical Staining"

_ijms, 2023, doi:10.3390/ijms24032926_

Round 1

Reviewer 1 Report

The authors have developed a well-conducted and well-written study with the aim of analyzing the expression of vascular endothelial growth factor and nerve growth factor in labra and their roles in osteoarthritis.

However, I suggest some clarifications or modifications that will in my opinion improve the quality of their manuscript:

1.In the introduction / Discussion section, I recommend the authors to comment on a possible relationship between the systemic origin of OA and the gut microbiota, mentioning the following reference: doi:10.3390/nu13030716.

2. The Discussion part should be further developed from the patient management point of view. I recommend the authors comment the following paper on the impact of hip OA from a systemic point of view: DOI: 10.1097/TGR.0000000000000337.

3. In the Discussion section, could you add a section on "Future Directions and Clinical Implications"?

Reviewer 2 Report

This study entitled “Expression of acetabular labral vascular endothelial growth factor and nerve growth factor is directly correlated with hip osteoarthritis pain” seems to have been generally well executed and written. Furthermore, I believe that this work will be of great interest to the readers. I have only a few minor suggestions to improve the quality of the paper.

Title

Please include the type of our study in your title. Furthermore, please replace the word correlated (it is more common to use in statistics) with a more suitable word (e.g., associated, related).

Introduction

It is more common to first state the aim of your study then the hypothesis of the study.

Materials and Methods

This section should be the second section of your paper, not the Results.

Please add in the first sentence what type of the study you have performed.

Results

Please write P values throughout your work in uniform way (i.e., three decimal places).

Reviewer 3 Report

Thank you for giving me the possibility to review the paper "Expression of acetabular labral vascular endothelial growth factor and nerve growth factor is directly correlated with hip osteoarthritis pain". This paper deals with a very interesting topic, since it aims to investigate the neurovascular etiology of pain and functional limitation in patients suffering for osteoarthritis.

However, before considering it for publication I IJMS, the following point should be addressed:

1. add a table that summarizes the sample data (age, gender, BMI, OA grading according to KL and Tonnis, the timing between hip pain onset and THA implantation, Harris Hip Score, NRS, comorbidities...)

2. please assess if there are any gender-related, age-related and BMI-related differences in NGF and VEGF in the recruited patients

3. please state if patients suffering from hip-spine syndrome were included or excluded from this study.

Round 2

Reviewer 1 Report

The authors have responded to all my suggestions.

I recommend their high quality work for publication.